# MolOptimizer: A Molecular Optimization Toolkit for Fragment-Based Drug Design

**DOI:** 10.3390/molecules29010276

**Published:** 2024-01-04

**Authors:** Adam Soffer, Samuel Joshua Viswas, Shahar Alon, Nofar Rozenberg, Amit Peled, Daniel Piro, Dan Vilenchik, Barak Akabayov

**Affiliations:** 1Department of Chemistry, Ben-Gurion University of the Negev, Beer-Sheva 8410501, Israel; 2Data Science Research Centre, Ben-Gurion University of the Negev, Beer-Sheva 8410501, Israel; 3Department of Software Engineering, Ben-Gurion University of the Negev, Beer-Sheva 8410501, Israel; 4School of Computer and Electrical Engineering, Ben-Gurion University of the Negev, Beer-Sheva 8410501, Israel

**Keywords:** cheminformatics, fragment screening, hit-to-lead optimization

## Abstract

MolOptimizer is a user-friendly computational toolkit designed to streamline the hit-to-lead optimization process in drug discovery. MolOptimizer extracts features and trains machine learning models using a user-provided, labeled, and small-molecule dataset to accurately predict the binding values of new small molecules that share similar scaffolds with the target in focus. Hosted on the Azure web-based server, MolOptimizer emerges as a vital resource, accelerating the discovery and development of novel drug candidates with improved binding properties.

## 1. Author Summary

The availability of advanced data aggregation, storage, labeling, and categorization tools has bridged the gap in hit-to-lead optimization. By leveraging data-driven algorithms, it is now possible to establish efficient and accurate design principles in drug discovery. Furthermore, efforts are being made to enhance the accessibility of these optimization tools to a broader user base.

Fragment-based screening is a prominent technique for identifying potential hit molecules from the vast chemical space. However, the subsequent “hit-to-lead” optimization step, which narrows the chemical space to achieve desired effects against a specific target, is extremely challenging. MolOptimizer is a toolkit designed for a hit-to-lead optimization of fragment-containing small molecules. The toolkit is available online (https://molopt.online/, accessed on 31 December 2023) or as an open-source (https://github.com/csbarak/MolOpt_Students_2023, accessed on 31 December 2023), and this paper provides a step-by-step guide on downloading, configuring, and utilizing the MolOptimizer toolkit.

## 2. Introduction

Fragment-based screening is a fundamental philosophy in drug discovery [1,2,3,4], which is based on finding fragment molecules that are bound to a macromolecular target. Small molecules in fragment libraries are only functional chemical groups. Therefore, optimizing a fragment hit into a drug-sized molecule and, at the same time, enhancing its affinity and specificity to the target is a highly challenging task. Due to their simplicity and small size (<300 Da), fragment molecules have a high propensity for target binding but often exhibit low affinity. Consequently, these fragment hits must be expanded into drug-sized molecules that increase attractive interactions with the pocket of the target macromolecule [4].

We have previously introduced a two-step computational process encompassing virtual filtration and virtual screening for a hit-to-lead optimization [4]. The subsets of fragment-containing drug-sized molecules selected from virtual libraries composed of multi-million drug-like small molecules underwent high-throughput molecular docking to obtain the binding value for each small molecule against a pre-structured drug target. The molecules were then ranked based on their calculated free energy binding to the target. The datasets of small molecules and their assigned molecular docking values (‘labels’) are crucial for developing data-driven prediction models that establish design principles for small molecules with enhanced binding properties. Specifically, we prepared a benchmark dataset of labeled small molecules obtained with NMR-fragment-based and virtual screening to target RNA hairpins [5]. The most influential chemical descriptors were employed to train machine learning models to predict the binding scores of novel small molecules (which contained the same scaffold found by the NMR fragment screening) [6].

In supervised learning, an algorithm learns from the labeled training data of small molecules, which is achieved using the features of the data to predict the labels on new, unseen data. The labels act as the ‘ground truth’ to guide the learning process. For a dataset of molecular descriptors, labels might represent the efficacy of binding to a specific target. By including these labels, the supervised learning model can discern the patterns and correlations between the features of the molecules (like shape, charge distribution, hydrophobicity, etc.) and their binding performance to the target.

Building on this knowledge, we present MolOptimizer, a toolkit that provides users with a simplified workflow for molecular optimization in fragment-based drug discovery. MolOptimizer integrates supervised learning models into a unified, user-friendly platform, which works to streamline the process and enhance the feasibility of molecular optimization. The unique features of MolOptimizer compared to other computational tools are presented in Appendix A.

## 3. Implementation

MolOptimizer is a Python-based computational tool devised to estimate the binding values for the chemical alterations that are made to pre-selected fragment molecules, thus eliminating the need for a time-consuming in silico docking process on a labeled subset. Details of MolOptimizer’s architecture can be found in the Appendix A. To facilitate ease of use, we offer a step-by-step video tutorial online, which guides users through molecular optimization using the benchmark dataset and MolOptimizer package provided (https://www.youtube.com/watch?v=2ouirsHHpJY, accessed on 31 December 2023). This workflow is delineated in a few straightforward steps, as illustrated in Figure 1.

Step 1: Upload a labeled dataset of small molecules, their binding scores, and a reference fragment molecule. Such a benchmark dataset, which features extracted descriptors for each molecule accompanied by a binding value (or label), is derived from AutoDock 4.2 [7] and Raccoon [8] tools (Figure 2), and it is provided with MolOptimizer.

Step 2: In this step, align a batch of small molecules using the atoms of the fragment molecule found in all of the entries in the dataset. This alignment maintains the structural orientation of the molecules and grants uniformity to the dataset. Utilize the RDKit’s Most Common Substructure (MCS) module (available at https://www.rdkit.org/, accessed on 31 December 2023) for this purpose, which operates based on the principles of tethered minimization (available at https://github.com/Discngine/rdkit_tethered_minimization, accessed on 31 December 2023). This module facilitates the identification of a common substructure by comparing the atoms and bonds between two molecules. Ensure you upload the reference molecule (the fragment hit) and a dataset of fragment-containing small molecules in SDF format. After alignment, the aligned dataset can be downloaded. MolOptimizer allows users to adjust the ‘ratio threshold’ to manage the size of the second dataset of fragment-containing drug-sized molecules while other default parameters remain constant.

Step 3: Extract a substantial number of chemical descriptors using the RDKit (https://www.rdkit.org/, accessed on 31 December 2023) and Mordred [12] Python libraries. These descriptors are critical for the computer-aided classification of molecules based on the structure–activity relationship (SAR) [13] (illustrated in Figure 3). Users can opt between RDKit or Mordred for feature extraction by uploading a multi-molecule file in .mol2 format. The extracted features are available for download in .csv format upon completion.

Step 4: Training machine learning models using the labeled datasets of small molecules to predict binding scores for new fragment-containing molecules. MolOptimizer features a user-friendly interface hosting three machine learning algorithms: extreme gradient boosting regression (XGBoost [14]), Lasso regression [15], and the decision tree regressor [16]. This phase enables the prediction of binding scores for new fragment-containing small molecules, thereby bypassing the necessity for molecular docking by considering the crucial chemical attributes embedded in the new entities.

The available data are split into two subsets to effectively train a machine learning model, i.e., the training and validation sets (where the split ratio is 75% and 25%, respectively). The training set is used to fit the model, thus allowing the algorithm to learn from the data. During this phase, a 5-fold cross-validation ensures the model’s performance is robust and not dependent on how the data are split. In a 5-fold cross-validation, the training set is divided into five smaller sets (folds). The model is trained five times, using four folds for training and the remaining fold for validation. This way, the model’s parameters are tuned, and the best model in terms of generalizing to new data is selected. After the model has been trained and the parameters have been selected, the separate validation set, which the model has not seen during the training process, is used to evaluate the model’s performance. This helps to provide an unbiased estimate of how well the model will perform on unseen data.

MolOptimizer offers machine learning algorithms in Expert Mode (Figure 4a) and Manual Mode (Figure 4b).

**Expert Mode:** In this setting, the MolOptimizer tool requires users to input specific hyperparameters that significantly influence the learning trajectory of the model. Designed to be user friendly, even for individuals with a limited background in machine learning, ‘Expert Mode’ automates the identification of optimal hyperparameters through the GridSearchCV function from the scikit-learn library [17]. Moreover, it suggests the top ten critical features using three integrated ML algorithms: XGBoost, Lasso regression, and the decision tree regressor.

The process involves training two models: a primary and a secondary one. Initially, the primary model is trained with the most influential features, which are identified automatically. Following this, the user re-uploads the dataset to train the secondary model, which utilizes the features selected from the recommended list. The model undergoes further refinement through another round of hyperparameter tuning with GridSearchCV before being saved as the final version for affinity score predictions.

Importantly, each step in our process creates files that the next step uses. For example, we line up molecules in a file in the molecular alignment step, ensuring they all match up in the same 3D space. Then, in the next part, we look at these files and pick out a great deal of different features for each molecule, as many as 1340 different kinds. The data are then organized into a tabular format for further analysis.

**Manual Mode:** This mode caters to users who prefer a hands-on approach, in which the option to input hyperparameters manually is offered, where chemical features are selected for training from the uploaded database, thus bypassing the extended training process witnessed in Expert Mode. It encompasses the functionalities from XGBoost, Lasso regression, and the decision tree regressor algorithms integrated within MolOptimizer.

To aid users in feature selection, which incorporates Shapley Additive exPlanations (SHAP [18]) for the XGBoost and the decision tree regressor, these notebooks assist in understanding the significant roles various features play in binding affinity predictions. Notably, SHAP is not implemented for Lasso regression due to current limitations. This framework, grounded in game theory, clarifies the model results by evaluating the impact of each feature on the predictions, thereby offering insights into the positive or negative influences on the outcome.

MolOptimizer incorporates various machine learning algorithms, including Extreme Gradient Boosting (XGBoost), Lasso regression, and the decision tree regressor. Here, we delve into each as follows:

**XGBoost:** The XGBoost algorithm [14], renowned for its tree-based learning and handling of large data volumes, is adept at predicting the binding affinity of drug-sized molecules. It is particularly beneficial in medicinal chemistry, addressing data sparsity issues by learning the direction of missing values, and in allowing tree splitting with sparsity awareness.

The aim of the XGBoost algorithm is minimizing the objective function, which is a combination of the loss function (L) and the regularization term (γ), which is as follows:(1)$${obj}^{\left(t\right)}=\sum_{i=1}^{n}\;\left[l\left(y_{i},{\overset{ˆ}{y}}_{i}^{\left(t-1\right)}\right)+g_{i}f_{t}\left(x_{i}\right)+\frac{1}{2}h_{i}f_{t}^{2}\left(x_{i}\right)\right]+\Omega \left(f_{t}\right)+constant,$$
where *g_i_* = ∂$\overset{ˆ}{y}$_(*t*−1)_l(*y_i_*,$\overset{ˆ}{y}$_i_^(*t*−1)^), *h_i_* = ∂^2^$\overset{ˆ}{y}$_(*t*−1)_l(*y_i_*,$\overset{ˆ}{y}$^(*t*−1)^), y is the target variable, and $\overset{ˆ}{y}$ is the predicted variable.

In MolOptimizer, users can utilize ‘Expert Mode’ to automatically find the optimal hyperparameters using GridSearchCV with a grid that includes parameters (with default values) such as ‘n_estimator’, ‘max_depth’, and ‘subsample’. Users can manually input hyperparameters like ‘Learning Rate’ and ‘Alpha Value’ in the ‘Manual Mode’ (Figure 5), whereby the model’s root mean square error displayed in the terminal.

**Lasso Regression:** This method, also known as shrinkage regression [15], minimizes a specific cost function and is crucial for reducing redundant features in large chemical datasets. It identifies and eliminates less relevant features, thus facilitating feature selection.

The cost function that the Lasso regression algorithm aims to minimize is as follows:(2)$$\sum_{k=1}^{n}\;{\left(y_{k}-\sum_{j}\;x_{kj}\beta_{j}\right)}^{2}+{\Theta \sum }_{j=1}^{p}\;\left|\beta_{j}\right|$$
where *y* is the target variable, *x* is the input variable, *β* is the magnitude, and Θ is the tuning parameter. The higher the value of Θ, the more the absolute value of coefficients shrinks, thus eliminating more features.

Users can either employ ‘Expert Mode’ to find the best regularization parameter using GridSearchCV or manually enter the desired value in ‘Manual Mode’.

A text file, ‘CustomModel_rmse.txt’, is generated in the root directory, which contains the mean absolute error, mean squared error, and root mean squared error of the model.

**Decision Tree Regressor:** In implementing the Classification and Regression Trees (CART) algorithm, this regressor predicts target variable values by continually splitting nodes to minimize the impurity via the Gini index that generalizes binomial variance [19]. ‘Expert Mode’ utilizes GridSearchCV to determine the optimal values for various hyperparameters, while ‘Manual Mode’ allows users to specify values for parameters like ‘max_depth’ and ‘min_samples_leaf’ (Figure 5). Creating a shallow decision tree is important for the simplification of the model.

For all three algorithms, the performance metrics such as the mean absolute error and root mean squared error are saved in a text file named ‘CustomModel_rmse.txt’ in the respective scripts’ sub-directories. Additionally, the SHAP approach detailed in the accompanying Jupyter Notebooks helps explain the tree structures created by the decision tree regressor.

## 4. Discussion

In this study, we presented a UI toolkit that seeks to simplify the hit-to-lead optimization process of a molecule that is obtained with fragment-based screening. Utilizing machine learning algorithms, the toolkit is designed to assist in analyzing a dataset that is uploaded by a user and which is aimed at identifying molecules with desired attributes more efficiently. The user’s dataset will contain molecules, and each of them are assigned a label of binding values that are obtained by virtual screening (docking).

MolOptimizer allows navigating the extensive chemical space more effectively, potentially reducing the time and resources traditionally required in the initial stages of drug discovery. Overall, the toolkit is available online (https://molopt.online/, accessed on 31 December 2023) and offers a practical approach to addressing the common challenges faced during the early phases of drug discovery, thereby facilitating a smoother transition from data analysis to pinpointing potential drug candidates.

## 5. Conclusions

This study introduces a toolkit that integrates machine learning algorithms to streamline fragment-based screening in drug discovery. By offering a more efficient pathway for analyzing datasets and identifying promising molecules, the tool serves as a simpler aid in the complex landscape of medicinal chemistry. The architecture of the toolkit and list of dependencies are presented in the Appendix A. It represents a thoughtful step toward refining the early stages of drug discovery, thus potentially making the process smoother and more focused. The tool has an available UI developed in Microsoft Azure^®^, and it can be downloaded from Github under a licensed agreement (https://github.com/csbarak/MolOpt_Students_2023, accessed on 31 December 2023). A video tutorial is available online at https://www.youtube.com/watch?v=2ouirsHHpJY, accessed on 31 December 2023.

## Figures and Tables

**Figure 1 molecules-29-00276-f001:**
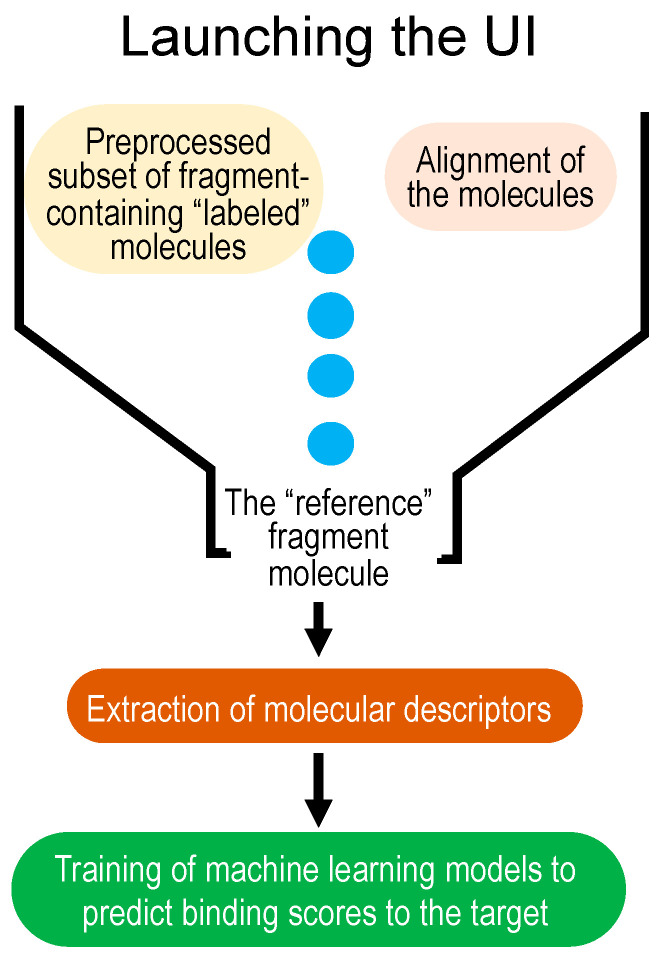
Workflow of MolOptimizer.

**Figure 2 molecules-29-00276-f002:**
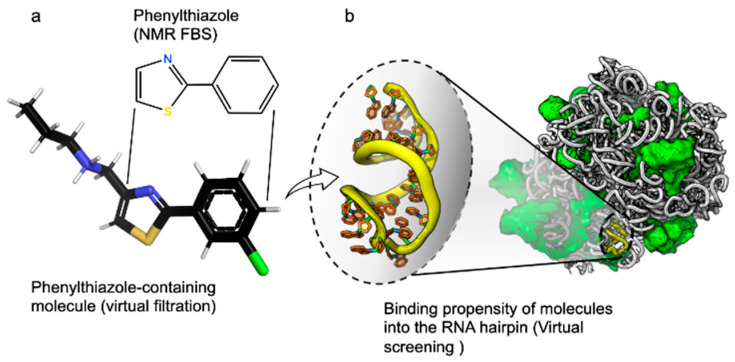
Benchmark dataset provided with MolOptimizer. The dataset contains binding values for 2-phenylthiazole-containing drug-sized molecules that bind RNA targets. (**a**) A molecular structure of 2-phenylthiazole, i.e., the scaffold in each molecule, was obtained by fragment-based screening using T2 relaxation spectroscopy [5]. A representative larger molecule containing phenylthiazole was obtained using a virtual filtration approach that was applied to the ZINC database [9]. (**b**) Hairpin 91, located in the center of the PTC of the large ribosomal subunit of *Staphylococcus aureus* (PDB id. 4WCE, [10]), was the target for the virtual screening of ~800 2-phenylthiazole containing small molecules. Molecular docking was performed using Autodock 4.2 [11].

**Figure 3 molecules-29-00276-f003:**
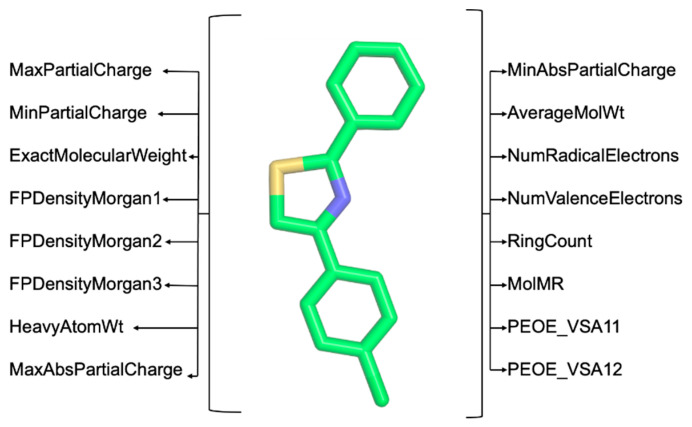
Example of chemical descriptors extracted using RDKit (https://www.rdkit.org/, accessed on 31 December 2023) and Mordred [12].

**Figure 4 molecules-29-00276-f004:**
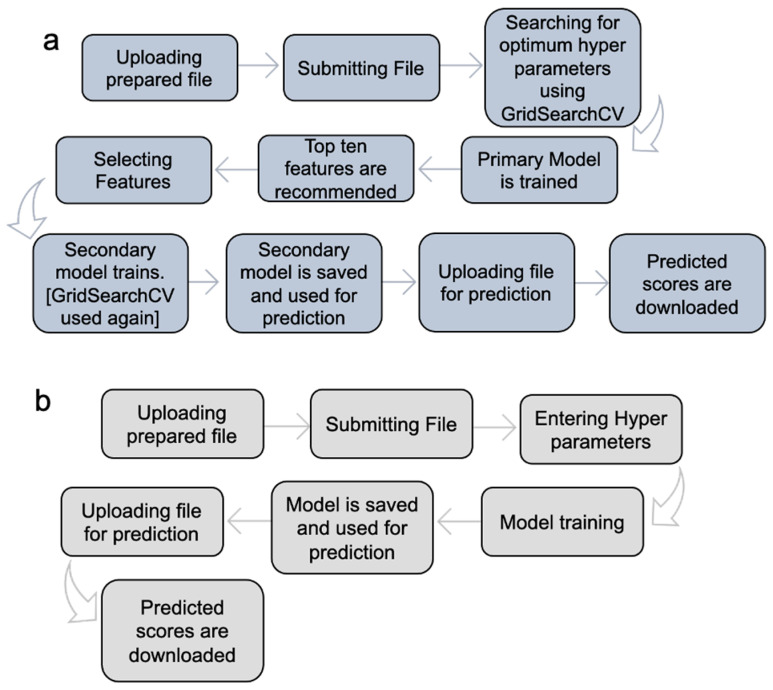
Implementation of MolOptimizer in Expert Mode (**a**) or Manual Mode (**b**). Expert Mode allows for the automatic selection of hyperparameters, whereas Manual Mode requires users to select the hyperparameters.

**Figure 5 molecules-29-00276-f005:**
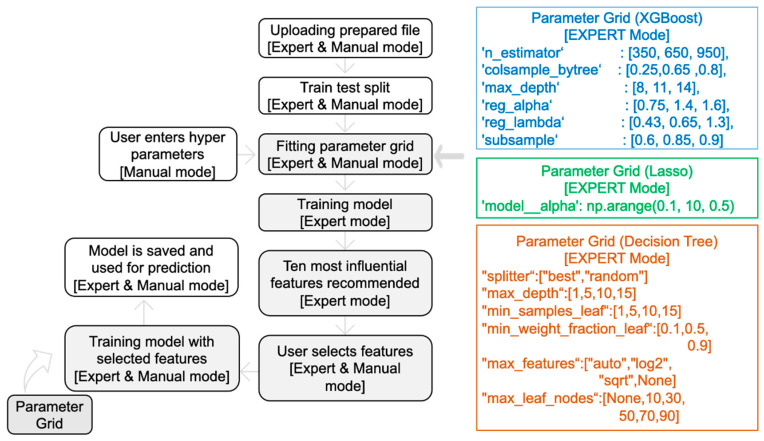
Training machine learning models with MolOptimizer.

## Data Availability

Project Name: MolOptimizer; Project home page: https://molopt.online/. Operating system: Platform-independent.

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
