# Peer review of "MolOptimizer: A Molecular Optimization Toolkit for Fragment-Based Drug Design"

_molecules, 2024, doi:10.3390/molecules29010276_

Round 1

Reviewer 1 Report

Comments and Suggestions for Authors

The authors present a new computational tool for fragment-based drug design. This is a very interesting approach since it allows a user to construct new molecules like a Lego puzzle. Although, not all parameters of potential candidates for drugs are taken into account, which are needed for estimation of solubility, bioavailability, etc., the developed toolkit is of potential interest and I’m pleased to recommend it for publication after a very minor revision. The methodology of fragment-based molecular search and screening is a milestone in designing of other special-purpose materials, in particular, energetic materials. Therefore, citation of the following paper and a short comment on the applicability of the fragment-based molecular search philosophy are highly suggested.

https://doi.org/10.1016/j.fpc.2023.07.002

Author Response

We thank the reviewers for the thoughtful review, which helped us improve the presentation of the reported MolOptimizer tool.

The file with all reviewers' comments is attached.

Reviewer 2 Report

Comments and Suggestions for Authors

This article from a team led by Barak Akabayov at Ben-Gurion University in Israel describes a "Toolkit" for optimizing fragment-based drug design.  The toolkit is for the most part composed of notable machine learning, etc. algorithms previously published by others, but have been combined here in a quite clever way.

The manuscript is probably publishable after some fairly minor revision, but major revisions could make it a much more interesting work than it is at present.  The major shortcoming is that there are no examples of the toolkit in action presented here.  It would be nice to see how it performs in 2-3 "classic" drug discovery scenarios, even as dated as the well-known discovery of HIV-1 protease inhibitors.  A side effect of this is that terminology such as in the abstract, "... emerges as a vital resource, accelerating the discovery ...", ring hollow without even tenuous evidence of it actually working.  Nevertheless, my opinion is that articles of this nature, that put forth a testable hypothesis or describe tools that can be used or replicated by other researchers, should be made available to the community, as long as the potential impact and value are not oversold.  I would prefer, as mentioned above, see some more proof of concept, however.

Comments on the Quality of English Language

English is generally OK.  There a handful of typos and awkward sentences that could be resolved by another pass of careful proofreading.

Author Response

(The authors gave the same response as above.)

Reviewer 3 Report

Comments and Suggestions for Authors

In this work, Soffer and co-workers present MolOptimizer, a software designed for calculation of the binding affinity of small drugs to macromolecules also known as molecular docking. The article provides a guide on how to use the software as well as a short description of the different features of the software. The development of open software is key for a more efficient global research in theoretical chemistry, therefore software-based papers as this work are important for the community. For this reason, I am positive about its publication as a tutorial in Molecules.

However, there are a few important points that the authors should consider to improve in the manuscript before its publication:

  1. What is the added value of MolOptimizer as compared to other already existing software for molecular software, for example GRID (Goodford 1985), POCKET (Levitt and Banaszak 1992), SURFNET (Laskowski 1995), PASS (Putative Active Sites with Spheres) (Brady and Stouten 2000), and MMC (mapping macromolecular topography) (Mezei 2003) and other software reviewed in 10.1007/s12551-016-0247-1 ? The authors should clearly present what are the unique features of their software as compared to other existing software.

  1. No description of how the software has been tested to ensure its reliability is provided. Against what benchmark have the authors validated their software? Do the authors have a set of test cases to ensure a trustworthy continuous integration for future development of the software?

  1. It is not clear to me what are the input files necessary to use MolOptimizer. Does it only accept the provided benchmark data set from AutoDock 4.2 and Raccoon? What is the expected format for the input files? On the same line,  the standard outputs of MolOptimizer should be more clearly stated.

  1. From the README file in the github repository of the software, I understand that MolOptimizer has several dependencies on other libraries. Such dependencies should be clearly stated in the paper, together with what specific version of each library.

  1. The authors provide a Youtube video as a complementary visual tutorial for the software. However, the video has no audio or subtitles which makes it hard to follow for people not familiar with the software.

Author Response

(The authors gave the same response as above.)

Round 2

Reviewer 3 Report

Comments and Suggestions for Authors

I thank the authors for addressing my suggestions and comments. In my opinion, the article is now publishable on its current form.